# Analyzing Factors Influencing Farmers in Northeast China to Convert from Corn to Rice Production

**Luan Wang [1], Chunmiao Liu [1] and Jing Zhang [2,*]**

[1] School of Economics, Shanxi University of Finance and Economics, Taiyuan 030006, China; luanwang@illinois.edu (L.W.); l17772531314@163.com (C.L.)

[2] Agricultural Information Institute, Chinese Academy of Agricultural Sciences, Beijing 100081, China

* Correspondence: zhangjing05@caas.cn; Tel.: +86-186-1056-0376

**Abstract:** The constantly changing prices of grains such as rice and corn have triggered an increasing number of corn-to-rice projects. This paper takes the progress of the corn-to-rice project in Northeast China as a sample and analyzes the influencing factors of the corn-to-rice project based on binary selection model analysis of the probit method. This study yields the following findings: the relative benefits of rice and corn are the key factors affecting the corn-to-rice project. When the project can improve farmers' income, their willingness to participate increases significantly; the number of farmers' family members providing labor is an important factor affecting the decision of corn-to-rice conversion. Accordingly, when the family labor is abundant, they are more willing to change from corn to rice. Farmers will choose the results that are beneficial to them according to their own conditions, and their choices can bring them greater economic benefits.

**Keywords:** corn to rice; farmers' participation behavior; binary selection model

## 1. Introduction

On the one hand, structural adjustment of the agricultural supply side and continuous development of the reform of the national grain price system has caused a steady reduction and sharp fluctuations in the price of corn. Corn prices in China fell 30% in 2017 compared to 2016 corn prices. The price of rice has been relatively high with stable output, showing the characteristics of better overall income. Compared with the price of rice in 2016, the price of rice increased by 15% in 2017 in China. On the other hand, the sown area and output of rice and corn have changed. In 2020, the sown area of rice was 30.076 million hectares, an increase of 1.29% year-on-year, and the output of rice was 212 million tons, an increase of 1.07% year-on-year. The sown area of corn was 41.264 million hectares, down by 0.05% year-on-year, and the output of corn was 261 million tons, down by 0.04% year-on-year. The price relation between corn and rice and the stable higher rice price promote the change from corn to rice, and such change will cause changes in the sown area and yield of rice and corn. As the main corn-producing area in the northeast, this trend is particularly active.

Changing from corn to rice has a greater impact on farmers' income, significantly increasing farmers' income [1]. Affected by natural conditions, the southern region has an earlier history of changing from corn to rice; accordingly, some experiences have been accumulated in the process [2,3]. The northeastern region had a late start in changing from corn to rice [4,5].

The government departments in Northeast China have seen the advantages of changing from corn to rice. Since 2012, they have encouraged farmers to participate in the corn-to-rice project in various areas. The corn-to-rice project refers to the development of water conservancy and irrigation, and the transformation of the original corn planting land into rice planting land. The corn-to-rice project can effectively improve the output efficiency of the land and form a stable production capacity.

In June–August 2019, the "Research on the Mechanism of 'Corn to Rice' and Agricultural Support Policy Innovation in Northeast China Based on the Structural Reform of Agricultural Supply Side" conducted a household survey in Northeast China. In Northeast China, no effective consensus exists on the policy of changing from corn to rice. The policies implemented in various regions are quite different. Tieling City and Xinmin City in Liaoning Province have given certain economic subsidies to farmers who have changed from corn to rice within a certain time limit. The project encourages farmers to convert from corn to rice. Minquan City and Harbin City built convenient irrigation measures for farmers from 2012 to 2017 to help farmers solve the problem of water use in the process of changing from corn to rice. Shuangyashan City is a traditional rice-growing area. It also shoulders the task of planting corn, thus providing no room for preferential policies. Instead, it encourages farmers to change from corn to rice according to their own wishes.

At the Central Rural Work Conference held in December 2020, President Xi Jinping emphasized the need to achieve a firm grasp of the initiative in food security and to speed up the yearly food production. The production concept of realizing basic self-sufficiency in food and ensuring national food security is a major strategic issue related to national economic development, social harmony, and the overall situation of China [6]. With the development of China's national economy and the improvement of people's living standards, people's demand for food has changed from "eat enough" to "eat well" [7]. As far as rice is concerned, 800 million people in the country rely on rice as their staple food, making rice the largest staple food in China [6]. During the "14th Five-Year Plan" period, China is facing the uncertainty of the "post-epidemic era" and Sino–US trade frictions; notably, the strategic security function of grain has seen further strengthening [8]. Seeing these demands, China is vigorously developing the rice planting industry. By 2020, the output of rice had increased by 51.2 million tons compared with 2003, contributing 21% to the overall increase in grain production [9].

The production of crops is characterized by the economies of scale [10]. When the level of agricultural production technology reaches a certain bottleneck, the scale of land planting becomes an important means of production [11,12]. The existing production and operation mode of small-scale farmers in China cannot meet the needs of future agricultural development [13]. To adjust the agricultural planting structure and meet the requirements of national food security, the interests of farmers should be taken into account [14]. Moreover, some notable obstacles exist in the process of large-scale agricultural production. First, with the development of industrialization and the acceleration of urbanization, more and more agricultural laborers, especially young and middle-aged rural laborers, have left the countryside and chosen non-agricultural employment with relatively high wages [15,16], and an increasing trend is observed in the aging and feminization of the labor force remaining in rural areas [17,18]. Therefore, the impact of agricultural labor shortage on agricultural production cannot be ignored. Second, agricultural inputs have decreased. Farmers who are continuing to cultivate the land have already begun to reduce their agricultural inputs and shift their agricultural income to the improvement of living standards. They no longer invest a certain percentage of agricultural income in more agricultural equipment to update production tools as before, but use most of their agricultural income to improve their lives. Farmers who are working in cities pay little attention to agricultural inputs. They put their main energy into areas other than agriculture, and many farmers do not even intend to return to the countryside in the future [19]. Third, the utilization rate of agricultural machinery still has room for improvement. The socialization of agricultural machinery services is relatively common, and the level of agricultural mechanization has been qualitatively improved. However, certain differences exist in the utilization rate of mechanization in the different stages of rice planting. In the process of seedling transplanting and rice drying, the utilization rate of agricultural machinery is low [20]. This finding is due to the lack of self-owned investment incentives, and the socialization of agricultural machinery services needs further improvement [21].

Affected by more people and less land, insufficient natural resource endowment, and China's land policy, smallholder production continues to be an important production and operation model in China [22]. The average operating scale of each household is less than 3.33 hectares, accounting for 97% of the total number of farmers [23]. This scenario leads to the problems of land fragmentation, a low level of mechanization, and high land transaction costs [24]. This situation also somewhat hinders the development of agriculture. Therefore, realizing large-scale operation through land transfer is worth exploring. In fact, as early as the 1980s, China put forward a call to gradually concentrate land with farming experts, which can effectively solve the problems of the fragmentation of agricultural land and low efficiency of small-scale land cultivation [25]. Scale expansion includes two forms: operation and land parcel-scale expansions [26]. The expansion of the decentralized land scale does not necessarily improve the economy of scale [27]. The expansion of the plot size is the basis for farmers to achieve economies of scale [28,29]. Previous studies have analyzed the realization process of land scale economy from the perspective of land circulation [30,31] but did not specifically analyze the agricultural planting willingness and the benefit. Based on our survey, this paper analyzes the factors that influence farmers to switch from corn to rice.

Most of the research on the rice and corn project is based on qualitative analysis, with less quantitative research. Since the scale of rice and corn projects in China is not large, and the farmers participating in the rice and corn projects are scattered, it is difficult to conduct quantitative analysis. Although these studies have given some explanations for the plight of the rice and corn project, they are not convincing enough.

This paper takes the progress of the corn-to-rice project for farmers in Northeast China as a sample and analyzes the influencing factors of the corn-to-rice project based on binary selection model analysis of the probit method. In order to better fulfill the purpose of this research, the structure of this paper is arranged as follows: the second part presents the model, introduces the specific analysis framework and model setting of this paper, and explains the variable selection; the third part presents research results, including model screening and analysis of regression results and a robustness test; the forth part is the discussion; and, finally, the fifth part gives the basic conclusions.

## 2. Materials and Methods

### 2.1. Empirical Model

In this paper, the probit model will be used to examine the influencing factors of farmers' participation behavior toward "corn-to-rice" projects in Northeast China. First, we assume that the dependent variable is a binary class variable. Therefore, the probit model can be used, and on this basis, the maximum likelihood estimation (MLE) method is used to conduct an empirical analysis of the model. Given that this paper uses panel data, the "panel binary selection model" can be used for analysis. For the respondents' choice behavior, that is, whether to change from corn to rice, a "latent variable" can be used to represent the net benefit of this choice behavior. The net income here refers to the comparison between the income from rice and corn. If the income from rice minus the income from corn is positive, this means that the net income is greater than 0; if the income from rice minus the income from corn is negative, this means that the net income is less than 0. If the net economic benefit of the respondents is greater than 0, as a rational person, they should choose to participate in this project; otherwise, they should not participate. We suppose the net income is

$$y_{it}^* = x_{it}'\beta + u_i + \varepsilon_{it}$$

where the net benefit represented by $y_{it}^*$ is an unobservable latent variable, $u_i$ is an individual effect, and $x_{it}$ is an explanatory variable.

Individual choices are

$$y_{it} = \begin{cases} 1 \text{ if } y_{it}^* > 0 \\ 0 \text{ if } y_{it}^* \leq 0 \end{cases}$$

$\varepsilon_{it}$ obeys the standard normal distribution; thus, given x, u, and β, the panel binary selection model that affects farmers' participation in the corn-to-rice project is

$$
\begin{aligned}
&P\left(y_{it}=1 \mid x_{it}, \beta, u_i\right) \\
&= P\left(y_{it}^{*}>0 \mid x_{it}, \beta, u_i\right) \\
&= P\left(x_{it}^{\prime}\beta + u_i + \varepsilon_{it}>0 \mid x_{it}, \beta, u_i\right) \\
&= P\left(\varepsilon_{it}>-u_i - x_{it}^{\prime}\beta \mid x_{it}, \beta, u_i\right) \\
&= P\left(\varepsilon_{it}<u_i + x_{it}^{\prime}\beta \mid x_{it}, \beta, u_i\right) \\
&= F\left(u_i + x_{it}^{\prime}\beta\right)
\end{aligned}
$$

where a is the cumulative distribution function of b, and it is necessary to assume that the density function of b is symmetrical on the origin. Y′ represents an unobservable latent variable; Y is an observable dependent variable, that is, the behavior of changing from corn to rice or not changing from corn to rice; X is an explanatory variable representing the planting income of the land [32,33], respondents' gender, education level [34], the proportion of household agricultural income, whether they are village cadres, income level, the number of household agricultural labor force participants, whether they have part-time work experience [35], whether the local government provides relevant training [36], household consumption expenditure, subsidies for rice planting, corn planting subsidies, and whether to join cooperatives, among others. Land planting income refers to the actual planting income per mu (one mu of land is equal to 666.67 square meters, and is an area measurement unit in China; the Chinese government generally provides subsidies based on the planting area per mu) of rice or corn planted on the land (the unit is RMB: Yuan), which can reflect the income of farmers from agriculture. The proportion of household agricultural income refers to the proportion of farmers' income from agriculture in their total income, which can reflect the degree of farmers' dependence on agriculture (unit: %). Income level position in the village refers to self-assessment of one's own income in the income position of the village. Whether there is any work experience refers to whether there is any experience of leaving the countryside and doing other work in other places, which can reflect the willingness of farmers to continue farming and explore other income sources. Whether the local government conducts relevant training refers to whether the local government has provided relevant equipment, irrigation measures, and planting course training for the corn-to-rice project. In order to promote the corn-to-rice project, governments in some areas will provide farmers with relevant training courses. Whether to join cooperatives refers to whether to join local agricultural cooperatives. Agricultural cooperatives can collectively purchase seeds, fertilizers, and other products, and find sales outlets for agricultural products together, which can help cooperative members obtain greater benefits. Household consumption expenditure refers to the overall expenditure of the family (the unit is RMB: yuan); here, we have calculated the family's food expenditure, clothing expenditure, communication expenditure, entertainment expenditure, medical expenditure, and other affairs expenditure. The corn planting subsidy and rice planting subsidy (the unit is RMB: yuan) is a certain amount of RMB subsidy given to farmers by the local government according to their planting area. Whether there are irrigation facilities refers to whether there are facilities and equipment that can provide irrigation for the field near the area. If there are related facilities, the stability of the corn yield can generally be greatly improved and provide a certain guarantee for farmers to obtain high yields. ε is the random disturbance term and u is the individual effect.

### 2.2. Research Data and Sample Characteristics

The data used in this paper are all from the household survey conducted by the research group of "Research on the Mechanism of 'Corn to Rice' in Northeast China and Agricultural Support Policy Innovation Based on the Structural Reform of Agricultural Supply Side". In 2017 and 2018, the project leader of the research group traveled to Northeast China for several inspections. From June to August 2019, he coordinated with the National Development and Reform Commission and the Ministry of Agriculture and then

selected Harbin and Shuangyashan in Heilongjiang Province for investigation. Shenyang City and Tieling City were selected for investigation in Liaoning Province. A total of 693 samples were obtained in the last visit. In the process of data collection, the researcher collected all the data at one time, and some data required the respondents to recall values from the past three years, which was very demanding for the respondents. Therefore, during data analysis, 54 samples were removed. The data used in the following analysis are from 639 samples. Table 1 shows the specific statistical characteristics of the data.

**Table 1.** Variable interpretation and assignment.

| Variable Name | Variable Code | Variable Assignment |
|---|---|---|
| Change from corn to rice | changecr | Changed from corn to rice assignment = 1; otherwise = 0 |
| Income from land cultivation | lnbenefit | Actual planting income |
| Gender | gender | 1 for males, 0 for females |
| Whether he is a village cadre | serv | 1 if he is a village cadre; otherwise, 0 |
| Education level | education | 1 for elementary school and below, 2 for junior high school, 3 for high school, 4 for college and above |
| Family farming workforce | labnum | Number of laborers actually working in agriculture |
| Do you have any working experience? | workout | 1 for yes, 0 for no |
| Share of family farming income | agriprop | 100% is 1; 80–99% is 2; 50–80% is 3; 30–50% is 4; below 30% is 5 |
| Income level position in the village | inclev | 1 for much lower; 2 for low; 3 for medium; 4 for higher; 5 for much higher |
| Household consumption expenditure | lnconsume | The total annual consumption expenditure of the whole family |
| Does the local government provide relevant training? | training | Participation in government-provided training is assigned a value of 1; otherwise, 0 |
| Subsidy for rice cultivation | ricesub | Amount of subsidy for rice cultivation per mu |
| Corn planting subsidy | cornsub | Amount of subsidy for corn planting per mu |
| Whether to join a cooperative | cooper | 1 for yes, 0 for no |
| Are there irrigation facilities near the cornfield? | cornirr | 1 if irrigation facilities exist; otherwise, 0 |

According to the statistical results in Table 2, 54% of the respondents in Northeast China have carried out corn-to-rice projects. In general, the proportion participating in corn-to-rice projects is relatively high compared with other corn-growing areas in China. The average income per mu of farming is RMB 944, but with certain fluctuations ranging from RMB 226 to RMB 2230. This fluctuation motivates farmers' willingness to change from corn into rice because compared with corn, the rice planting process relies more on agricultural machinery irrigation, and is less affected by the external environment and has a relatively stable output. Less fluctuation in the purchase price of rice leads to predictable income. However, compared to rice, corn is more affected by external environments such as weather, causing a highly fluctuating yield. (On the one hand, the unit price of corn is relatively low, and farmers are less willing to build irrigation facilities for corn planting. On the other hand, Northeast China has relatively abundant rainfall, so there is little demand for irrigation facilities. Therefore, for corn-planting areas, the penetration rate of irrigation facilities is low. This has also caused the yield of corn to be greatly affected and the income of planting corn to fluctuate greatly when the natural conditions are not good.)

**Table 2.** Basic results from the survey of the main variables.

| Variable | Obs | Mean | Std. Dev. | Min | Max |
|---|---|---|---|---|---|
| changecr | 639 | 0.54 | 0.50 | 0.00 | 1.00 |
| lnbenefit | 639 | 6.85 | 0.42 | 5.42 | 7.71 |
| gender | 639 | 0.92 | 0.28 | 0.00 | 1.00 |
| serv | 639 | 0.15 | 0.35 | 0.00 | 1.00 |
| education | 639 | 1.88 | 0.73 | 1.00 | 4.00 |
| labnum | 639 | 2.90 | 1.21 | 1.00 | 7.00 |
| workout | 639 | 0.41 | 0.49 | 0.00 | 1.00 |
| agriprop | 639 | 2.01 | 1.15 | 1.00 | 5.00 |
| inclev | 639 | 2.82 | 0.80 | 1.00 | 5.00 |
| lnconsume | 639 | 10.48 | 0.66 | 8.52 | 11.58 |
| training | 639 | 0.45 | 0.50 | 0.00 | 1.00 |
| ricesub | 639 | 63.77 | 56.39 | 0.00 | 212.00 |
| cornsub | 639 | 121.95 | 80.85 | 0.00 | 248.01 |
| cooper | 639 | 0.24 | 0.43 | 0.00 | 1.00 |
| cornirr | 639 | 0.33 | 0.47 | 0 | 1 |

In terms of personal characteristics, 186 farmers received primary school education or below, accounting for 29%; 372 farmers received junior high school education, accounting for 58%; 54 farmers received high school education, accounting for 9%; and only 27 individuals received college education or above, accounting for 4%. The educational level of farmers is generally low. Farmers with part-time work experience accounted for 41%, and the overall proportion is not high. With the increase in the income level of farmers, farmers in the Northeast region rely more on land for income growth, and their income is relatively weak.

In terms of the state's training for farmers, 45% of farmers have received training from the Chinese government, which shows that the northeastern government attaches high importance to "the three rural issues" and has achieved certain results in terms of improving farmers' agricultural skills. (The "Three Rural Issues" refers to China's agriculture, rural areas, and farmers. The annual statistical report of the National Bureau of Statistics of China shows that by the end of 2021, there will still be 498 million farmers in China, accounting for 35% of China's total population. Therefore, agriculture, rural areas, and farmers' issues are very important in China).

## 3. Research Results

### 3.1. Model Screening

For an accurate selection of a model, we conducted screening in two ways. The models obtained through the screening can help us examine the factors that influence farmers' decision to participate in the corn-to-rice project. First, as a control, OLS was selected for linear probability model (LPM) estimation. Second, from the results of the logit model and rlogit model, the robust standard error is extremely close to the ordinary standard error, thereby eliminating the need to consider the problem of model setting. Finally, comparing the calculations of the rlogit model and probit model, the marginal effect, quasi R2, and correct prediction ratio of the logit model are very similar to those of the probit model. Thus, they can be regarded as equivalent. Therefore, the choice of the probit model in this paper should be theoretically correct. As shown in Table 3, we analyzed the regression results of the OLS model, the logit model, the robust logit model, and the panel probit model. The results show that the regression results of the four models only have a certain difference in the size of the coefficients, and no obvious difference exists in the significance and direction of the coefficients. Given that the data are in the form of panel data, the following analysis will be based on the probit model.

**Table 3.** Model regression results.

|  | OLS | Logit | Rlogit | Probit |
|---|---|---|---|---|
| lnbenefit | 0.614 *** | 3.754 *** | 3.754 *** | 2.156 *** |
|  | (14.81) | (11.2) | (10.23) | (12.19) |
| gender | −0.061 | −0.409 | −0.409 | −0.174 |
|  | (−1.14) | (−1.01) | (−1.16) | (−0.74) |
| serv | −0.094 | −0.709 ** | −0.709 * | −0.385 * |
|  | (−1.55) | (−2.04) | (−1.87) | (−1.95) |
| education | 0.027 | 0.216 | 0.216 | 0.113 |
|  | (1.1) | (1.31) | (1.33) | (1.18) |
| labnum | 0.036 *** | 0.236 ** | 0.236 ** | 0.136 ** |
|  | (2.78) | (2.42) | (2.48) | (2.43) |
| workout | 0.077 ** | 0.563 ** | 0.563 ** | 0.326 *** |
|  | (2.25) | (2.57) | (2.51) | (2.58) |
| agriprop | −0.038 ** | −0.253 ** | −0.253 ** | −0.146 ** |
|  | (−2.40) | (−2.50) | (−2.42) | (−2.52) |
| inclev | 0.069 *** | 0.492 *** | 0.492 *** | 0.288 *** |
|  | (3.45) | (3.3) | (3.38) | (3.35) |
| lnconsume | −0.005 | −0.072 | −0.072 | −0.017 |
|  | (−0.20) | (−0.41) | (−0.40) | (−0.16) |
| training | −0.086 *** | −0.522 ** | −0.522 ** | −0.311 ** |
|  | (−2.67) | (−2.37) | (−2.49) | (−2.42) |
| ricesub | −0.001 *** | −0.009 *** | −0.009 *** | −0.005 *** |
|  | (−4.62) | (−4.24) | (−4.11) | (−4.14) |
| cornsub | −0.001 *** | −0.007 *** | −0.007 *** | −0.004 *** |
|  | (−4.97) | (−4.96) | (−4.65) | (−4.98) |
| cooper | 0.072 * | 0.416 | 0.416 | 0.257 * |
|  | (−1.68) | (−1.56) | (1.55) | (−1.65) |
| cornirr | −0.084 ** | −0.474 ** | −0.474 ** | −0.285 ** |
|  | (−2.34) | (−1.99) | (−2.04) | (−2.07) |
| _cons | −3.544 *** | −24.581 *** | −24.581 *** | −14.408 *** |
|  | (−9.36) | (−8.65) | (−8.76) | (−9.05) |
| N | 639 | 639 | 639 | 639 |

* $p < 0.1$, ** $p < 0.05$, *** $p < 0.01$.

Before performing regression analysis on the data, we first calculated the correlation coefficient matrix and expansion factor of each variable to judge whether the variables used have collinearity problems. From the correlation coefficient matrix in Table A2 in Appendix A, the coefficients between the variables are generally less than 0.3. From the results of the inflation factor in Table A1 in Appendix A, the inflation factors between the variables are all between 1 and 2, and the average inflation factor is 1.35. Therefore, there are no serious collinearity problems between the variables used in this paper.

*3.2. Analysis of Regression Results*

Table 3 shows the model estimation results. The regression analysis of the main variables is consistent with the discussion, and the specific analysis is as follows.

The benefit of participating in the corn-to-rice project is significantly positively correlated to farmers' decision to change from corn to rice. This finding shows that the benefits of corn-to-rice conversion are the main determinants of whether farmers will undertake corn-to-rice projects. From an economic point of view, every farmer is a rational person, and their decisions are closely related to their interests. The greater the benefit of the corn-to-rice project, the more motivated they will be to carry out the corn-to-rice program, and the more likely they will be to implement the latter.

A village cadre (serv) has a significant negative impact on farmers changing from corn to rice, which indicates that village cadres are less willing to change from corn to rice. Village cadres generally have some business affairs to handle, which will take up a certain amount of labor time. Moreover, after becoming a village cadre, they will gain other income

apart from farming, which will also compel the village cadres to shift part of their energy from agriculture to their other work. Specifically, given that village cadres are engaged in work other than agriculture, they will reduce the time spent working in agriculture. In addition, the degree of mechanization of planting corn is significantly higher than that of rice, essentially realizing the entire mechanization process. This situation requires less time for individuals engaged in agricultural labor. Due to an increased demand for labor time for growing rice, compared to corn, the labor demand for village cadres is clearly higher. Therefore, for time reasons, the willingness of village cadres to change from corn to rice is significantly reduced.

The number of household laborers (labnum) has a significant positive impact on farmers' corn-to-rice programs. The corn-to-rice project requires a substantial amount of human input. Although part of the rice planting work can be mechanized, part of the rice planting work still needs manual operation. If the number of household laborers is small, laborers must be hired to complete the work, which will cause obvious substantial reductions in the profit level of agricultural cultivation. Therefore, the number of family laborers is found to have a significant role in promoting the corn-to-rice project.

Irrigation facilities near corn and other crops have a significant negative impact on farmers' conversion in corn-to-rice projects. The sunk cost of irrigation facilities is relatively high, and irrigation facilities can help corn be obtained in stable high yields. When irrigation facilities are built around cornfields, farmers are often unwilling to change their farming methods and change the existing cornfields to rice fields. Corn fields where irrigation facilities can be built are generally more fertile. The purpose of building irrigation facilities is to increase production and maintain sustainability. Since the sustainability of corn is better, farmers are less willing to change crop varieties. Therefore, irrigation facilities in cornfields have a significant negative impact on farmers' conversion in corn-to-rice projects.

*3.3. Robustness Test*

3.3.1. Hausman Test

The result of the Hausman test is

$$\text{Chi2}(13) = (b - B)'\,(v_b - v_B)^{-1}\,(b - B) = 126.22$$

As seen from the results in Table 4, the mixed probit regression is strongly rejected, thereby requiring the use of a random effects panel probit estimate. From the regression results of the random effects panel probit model, it is similar to the random effects panel logit model. Therefore, finally choosing the regression of the random effects panel probit model is accurate.

**Table 4.** Hausman test.

| | **(b)** | **(B)** | **(b-B)** | **Sqrt [diag($v_b - v_B$)]** |
|---|---|---|---|---|
| | RE | POOLED | Difference | S.E. |
| lnbenefit | 4.292 | 2.132 | 2.160 | 1.425 |
| gender | 2.586 | (0.189) | 2.775 | 1.740 |
| serv | (3.449) | (0.408) | (3.041) | 1.846 |
| education | 0.885 | 0.107 | 0.778 | 0.691 |
| labnum | 1.782 | 0.132 | 1.651 | 0.434 |
| workout | 4.307 | 0.356 | 3.952 | 0.980 |
| agriprop | (1.800) | (0.156) | (1.644) | 0.562 |
| inclev | 2.768 | 0.261 | 2.507 | 0.785 |
| lnconsume | (0.688) | (0.014) | (0.673) | 0.702 |
| training | (0.248) | (0.278) | 0.029 | 1.035 |
| ricesub | (0.022) | (0.005) | (0.017) | 0.008 |
| drysub | (0.014) | (0.004) | (0.010) | 0.006 |
| cooper | 2.763 | 0.261 | 2.502 | 2.194 |

### 3.3.2. Individual Heterogeneity Test

The standard probit model generally assumes that the random disturbance term is homoscedastic, and its null hypothesis $H_0$ is

$$P(y_i = 1 | x_i) = \varphi\left(\frac{x_i'\beta}{\sigma}\right). \tag{1}$$

where the standard deviation of the disturbance term is $\sigma = 1$. In addition, the assumption of heteroscedasticity $H_1$ is

$$P(y_i = 1 | x_i) = \varphi\left(\frac{x_i'\beta}{\sigma}\right). \tag{2}$$

However, $\sigma_i^2 = Var(\varepsilon_i)$. Then, it can be assumed that $\sigma_i^2$ depends on the exogenous variable $z = (z_1, z_2, z_3, \ldots \ldots, z_n)$. Therefore, we can obtain

$$\sigma_i^2 = \exp(z_i'\delta). \tag{3}$$

The exogenous variable z here can overlap with the independent variable. Taking the logarithm of both sides of Equation (3), we obtain

$$ln\sigma_i^2 = z_i'\delta \tag{4}$$

Under the alternative assumption of heteroscedasticity, the likelihood function can also be written, estimating both Equations (2) and (4).

Stata was used to estimate the results. The latter showed that the $p$ value of the likelihood ratio test was 0.21. Therefore, the test of homoscedasticity was acceptable.

### 3.3.3. Model Prediction Accuracy Test

Third, the prediction accuracy of the random panel probit model was analyzed, and the prediction accuracy of the panel probit model was found to be 77.62%. The model has high prediction accuracy and can effectively describe the survey data.

### 3.3.4. Influence of Relevant Characteristics on the Behavior of Changing from Corn to Rice

Figures 1 and 2 can more intuitively reflect the impact of the number of family laborers, whether they are village cadres, and whether they have part-time work experience on farmers' behavior of changing from corn to rice. As can be seen from Figure 1, with the increase in the number of household laborers, the probability of farmers switching from corn to rice increases significantly. The enthusiasm of village cadres to change from corn to rice is not high, and the probability of full-time farmers choosing corn over rice has increased significantly in the past. Figure 2 also shows that farmers with part-time work experience are more inclined to undertake corn-to-rice projects.

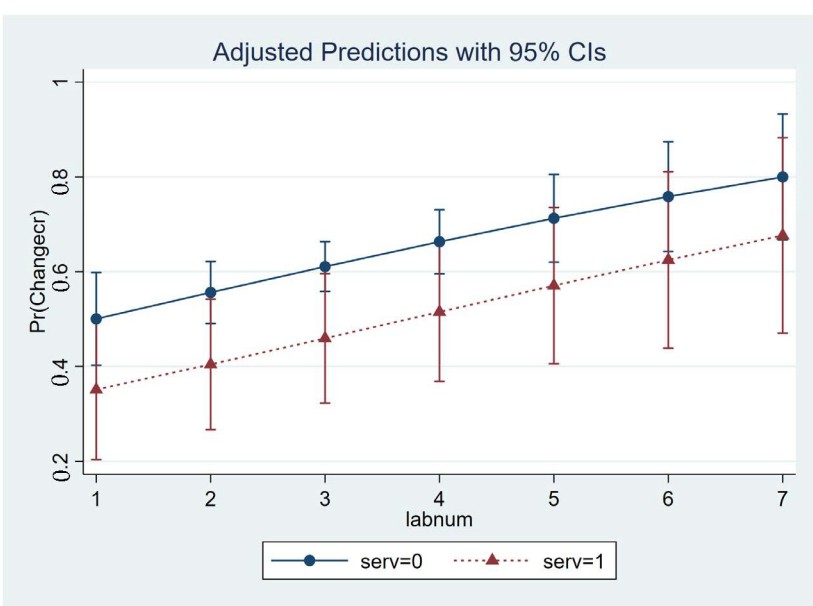

**Figure 1.** Effect of being a village cadre (serv = 1) on changing from corn to rice.

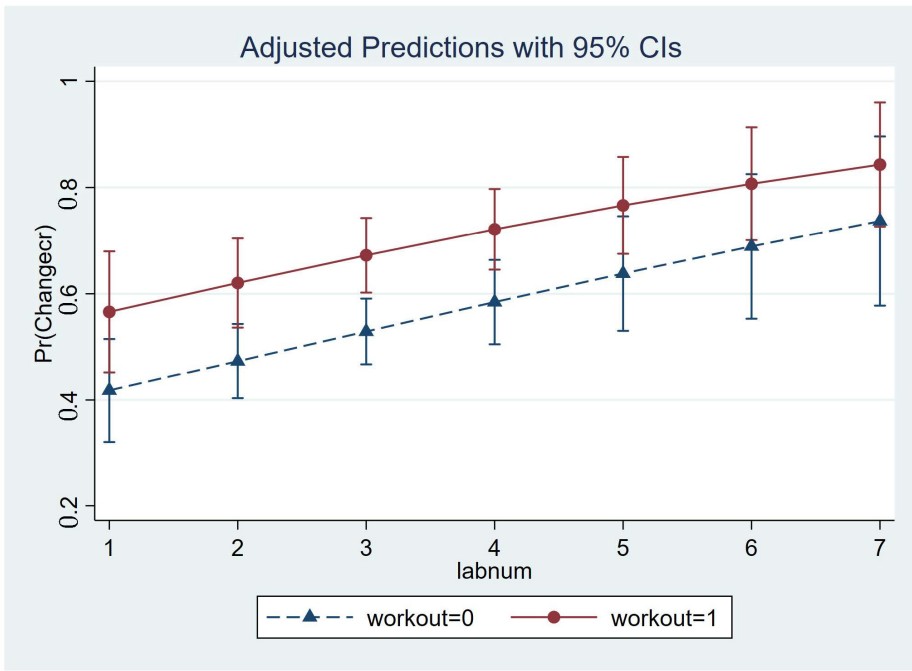

**Figure 2.** Influence of work experience (workout = 1) on changing from corn to rice.

## 4. Discussion

Table 5 presents the basic statistical results of this paper. With regard to the gender of the farmers, among the 639 respondents, only 54 were women and 585 were men. However, only 33% of women were involved in corn-to-rice projects, compared with 56% of households with men as the main labor force because the conversion from corn to rice is more physically demanding work, and operating the latter is difficult for women. The family's workload substantially increases after switching from corn to rice, which is more difficult for families with women as the main labor force.

**Table 5.** Statistical relationship analysis of individual characteristics in changing from corn to rice.

| | Did Not Change from Corn to Rice | Changed from Corn to Rice | Total |
|---|---|---|---|
| Gender | | | |
| Female | 36 | 18 | 54 |
| Male | 255 | 330 | 585 |
| Whether he is a village cadre | | | |
| Not a village cadre | 243 | 302 | 545 |
| Is a village cadre | 48 | 46 | 94 |
| Do you have working experience | | | |
| No working experience | 189 | 191 | 380 |
| Have working experience | 102 | 157 | 259 |
| Have you participated in agricultural training? | | | |
| No agricultural training | 151 | 201 | 352 |
| Participated in agricultural training | 140 | 147 | 287 |

From the data, the following scenarios can be seen: whether or not they are employed in the village, the proportion of village cadres who carry out corn-to-rice projects is roughly 16%, and the proportion of farmers who are not village cadres is as high as 47%. This finding shows that the proportion of village cadres who change from corn to rice is much lower than that of villagers who change from corn to rice. During the investigation, we learned that this finding is due to the fact that the demand of work for crop cultivation will be highly increased when the crop is changed from to rice, and many village cadres lack sufficient time to manage the farmland. Moreover, village cadres no longer use farming as their main source of income. Consequently, they do not prioritize spending time on land farming.

From the perspective of whether they had experience of leaving the countryside and doing other work in other places, 259 farmers had experience of leaving the countryside and doing other work in other places, accounting for 40% of all farmers. For farmers with working experience, 41.7% participated in the corn-to-rice project. For the farmers who had no experience of going out to work, the proportion of those participating in the corn-to-rice project was 45%. Farmers without part-time work experience had a higher proportion of corn-to-rice projects than those with part-time work experience. This finding is due to the fact that farmers who have no working experience tend to make a living through farming, and agricultural income accounts for a larger proportion of their total income. Notably, the income from rice tends to be higher than from corn (because the only source of income for these farmers who have no experience of going out to work is agriculture, if they want to increase their income, they can only grow rice, so they tend to change from corn to rice). Therefore, farmers who have no other source of income are more willing to participate in corn-to-rice projects.

A total of 287 farmers participated in the agricultural training organized by the government, accounting for 44.91% of the total farmers. Among those who participated in the training, the proportion of corn-to-rice projects was 43.9%. The proportion of farmers who had not participated in the training on the corn-to-rice project was 43.47%.

The corn-to-rice project is a project encouraged by the Chinese local government and voluntarily participated in by farmers. In terms of whether to participate in the project, the government guides farmers to participate by constructing irrigation facilities and distributing subsidies. When farmers participate in the corn-to-rice project, they can

convert part of their corn land into rice land, or convert all of their corn land into rice land, depending on the farmers' own wishes. All the data in this study come from the survey data of the Chinese social science project "Research on the Mechanism of Changing Corn to Rice in Northeast China Based on Agricultural Supply-side Structural Reform and Agricultural Support Policy Innovation", which is limited by the amount of data, quality, and availability, so there are some analyses that we cannot effectively perform. In this study, we mainly studied the influencing factors of whether to participate in the corn-to-rice project, but the analysis of the sowing area of different varieties and the input data of different crops is insufficient, which is the direction of our next research.

## 5. Conclusions

Farmers' decisions to participate in a corn-to-rice-project are mainly connected to the possibility to increase their income due to higher prices for rice than for corn. From the research of this paper, whether farmers participate in the corn-to-rice project is found to be affected by the benefits. If the corn-to-rice project can increase agricultural income, this seems to be the main driver; if farmers cannot obtain economic benefits from the corn-to-rice project, their motivation to participate in the project will be significantly reduced. Other reasons are as follows:

Whether or not he is a village cadre is an important factor affecting farmers' participation in the corn-to-rice project. Given that rice planting requires additional labor time investment and with less labor time of village cadres, this will reduce their willingness to change from corn to rice.

The number of household laborers will also affect the decision of farmers to participate in the corn-to-rice project. The pre-transformation process of changing from corn to rice and the planting process after changing from corn to rice have higher demands on the labor force. Agricultural returns suffer when households are short of labor and rely on hiring to maintain rice cultivation.

The above conclusions have important policy implications for the corn-to-rice project:

From the perspective of individual farmers, participation in the corn-to-rice project is restricted by many factors, and farmers must choose according to their own circumstances. First of all, predicting the profitability of the project is necessary. If the corn-to-rice project can increase farmers' income, the project will be more beneficial. Second, the farmers' family situation needs consideration. Compared with corn planting, the project of changing from corn to rice requires a higher number of farm family members.

From the government's point of view, the government should recognize that the corn-to-rice project is a market economy activity. Farmers will make rational choices according to their own circumstances. If the corn-to-rice project can bring benefits to farmers, the enthusiasm of farmers to participate in the corn-to-rice project will increase. Otherwise, it will decrease. As farmers better understand how to farm, the government should not participate too much in the specific operation of the project nor should it use too many factors such as policies and funds to intervene in market economic behavior.

**Author Contributions:** L.W.: Investigation, questionnaire collation, empirical analysis, formal analysis, and writing—original draft. C.L.: Data entry, data search, and document review. J.Z.: Project leader, supervision, validation, and corresponding author. All authors have read and agreed to the published version of the manuscript.

**Funding:** This article was sponsored by the Research on the Mechanism of "Corn to Rice" in the Northeast Region Based on the Structural Reform of the Agricultural Supply Side and the Innovation of Agricultural Support Policy (grant number: 17CJY033) and Research on the Equilibrium Wage and Its Impact Realized by the New Generation of farmers who work outside Through Bargaining under the Background of Incomplete Information" (grant number: 2019W080).

**Institutional Review Board Statement:** Not applicable.

**Informed Consent Statement:** Informed consent was obtained from all subjects involved in the study.

**Data Availability Statement:** The data of this study come from the survey conducted by the research group in 2019.

**Acknowledgments:** This is our first English submission. The author is very grateful to the editor and anonymous reviewers for their professional opinions and suggestions. These opinions and suggestions are very useful for us to revise the paper and let us learn a lot in the revision of the paper. The authors would also like to thank Jing Zhang for his great help with this article.

**Conflicts of Interest:** The authors declare no conflict of interest.

## Appendix A

**Table A1.** VIF table.

| Variable | VIF | 1/VIF |
| --- | --- | --- |
| lnarea | 2.24 | 0.45 |
| agriprop | 1.52 | 0.66 |
| lnconsume | 1.49 | 0.67 |
| serv | 1.49 | 0.67 |
| drysub | 1.43 | 0.70 |
| ricesub | 1.40 | 0.71 |
| education | 1.33 | 0.75 |
| cooper | 1.27 | 0.79 |
| inclev | 1.21 | 0.82 |
| lnbenefit | 1.20 | 0.84 |
| gender | 1.19 | 0.84 |
| labnum | 1.16 | 0.86 |
| dryirr | 1.11 | 0.90 |
| training | 1.11 | 0.90 |
| workout | 1.10 | 0.91 |
| Mean VIF | 1.35 | |

**Table A2.** Correlation table.

| | Changecr | Lnbenefit | Gender | Serv | Education | Labnum | Workout | Agriprop | Inclev | Serv | Lnconsume | Training | Ricesub | Drysub | Cooper | Dryirr | Lnarea |
| --- | --- | --- | --- | --- | --- | --- | --- | --- | --- | --- | --- | --- | --- | --- | --- | --- | --- |
| changecr | 1 | | | | | | | | | | | | | | | | |
| lnbenefit | 0.4614 * | 1 | | | | | | | | | | | | | | | |
| gender | 0.1289 * | 0.1126 * | 1 | | | | | | | | | | | | | | |
| serv | −0.0461 | 0.0892 * | −0.0644 | 1 | | | | | | | | | | | | | |
| education | 0.0755 | 0.0786 * | 0.0418 | 0.4157 * | 1 | | | | | | | | | | | | |
| labnum | 0.1293 * | 0.0999 * | −0.0119 | −0.012 | 0.0389 | 1 | | | | | | | | | | | |
| workout | 0.1021 * | 0.0982 * | 0.0102 | 0.1431 * | 0.0596 | 0.1543 * | 1 | | | | | | | | | | |
| agriprop | −0.1460 * | 0.0539 | −0.2036 * | 0.2741 * | 0.1308 * | 0.0378 | 0.1740 * | 1 | | | | | | | | | |
| inclev | 0.1778 * | 0.1543 * | 0.0803 * | 0.1595 * | 0.1891 * | 0.0780 * | 0.0288 | −0.029 | 1 | | | | | | | | |
| serv | −0.0461 | 0.0892 * | −0.0644 | 1.0000 * | 0.4157 * | −0.012 | 0.1431 * | 0.2741 * | 0.1595 * | 1 | | | | | | | |
| lnconsume | 0.1223 * | −0.041 | 0.0840 * | 0.0173 | 0.2157 * | 0.2400 * | 0.0608 | −0.1347 * | 0.1389 * | 0.017 | 1 | | | | | | |
| training | −0.0588 | 0.0023 | −0.0311 | 0.1668 * | 0.1428 * | −0.0631 | −0.0213 | −0.0184 | 0.1943 * | 0.1668 * | 0.0747 | 1 | | | | | |
| ricesub | −0.2380 * | 0.073 | −0.1758 * | 0.1701 * | −0.0606 | −0.0162 | 0.019 | 0.2759 * | −0.0208 | 0.1701 * | −0.3102 * | −0.0004 | 1 | | | | |
| drysub | −0.1011 * | 0.3323 * | −0.1913 * | 0.0761 | −0.0362 | 0.1316 * | 0.0755 | 0.2014 * | −0.0129 | 0.076 | −0.2112 * | −0.0145 | 0.3751 * | 1 | | | |
| cooper | 0.0607 | 0.0276 | 0.0905 * | 0.3594 * | 0.2604 * | −0.0766 | 0.0553 | 0.0371 | 0.2358 * | 0.3594 * | 0.0895 * | 0.2345 * | −0.006 | −0.0494 | 1 | | |
| dryirr | −0.006 | 0.1230 * | 0.1571 * | 0.0946 * | 0.0972 * | 0.1186 * | 0.0535 | 0.0584 | 0.1431 * | 0.0946 * | 0.0175 | 0.022 | 0.0125 | 0.0949 * | 0.0264 | 1 | |
| lnarea | 0.2417 * | −0.065 | 0.2826 * | −0.0941 * | 0.0645 | 0.0698 | −0.1384 * | −0.5117 * | 0.2272 * | −0.0941 * | 0.4683 * | 0.0667 | −0.4500 * | −0.4107 * | 0.1154 * | −0.0646 | 1 |

* $p < 0.05$.

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
