# Peer review of "Analyzing Factors Influencing Farmers in Northeast China to Convert from Corn to Rice Production"

_sustainability, doi:10.3390/su151310120_

Round 1

Reviewer 1 Report

This paper attempts to analyze the factors that influence "Corn to Rice" Farmers' Participation Behavior in Northeast China. The introduction does not provide a good background for the study and should be expanded significantly. The literature review section attempts to report previous work rather than providing a critical review. I have some concerns about the methodology. Some variables, for example, are categorical variables that can be converted to dummy variables. Otherwise, meaningful regression coefficient results cannot be obtained (education level, Income level position in the village). It would be prudent to investigate the variables' correlation. Furthermore, it is unclear to me what the OLS model's dependent variable is. The economic interpretation of the results must be clearly provided. Include any potential policy implications in the conclusion.

Author Response

Thank you very much for your suggestion, we have made the following modifications according to your suggestion. Please see the attachment.

Reviewer 2 Report

The manuscript by Authors are an interesting and relevant appraisal; and have the potential to attract the readers and researchers. Authors put the effort on writing of review and discussion. However, there are some aspects that need to be addressed in the revised copy.

1.       Abstract: There is no need any improvement in this section and keep the format of keywords per guidelines.

3.       Introduction: Well written

3.       Material and methods:

·       Author use OLS, logit, rlogit and probit model in the results and discussion section, but only explains probit model in detail.

·       Criteria to choose the probit model is not very clear

·       Selection of variable code is inappropriate in some cases such as change from corn to rice, whether he is a village cadre etc.

·       Title of Table 3 is not match with content. Table 3 explains the frequency distribution of categorical variable such as gender, working experience etc. Unit of the table is also not so clear.

4. Results and discussion:

·       Provide the details of table of correlation matrix and variance inflation factor (VIF) in Annexure.

·       name of variable use is very lengthy and difficult to retain while reading the article

·       R2 and Adjusted R2 in place of code given for the same in Table 4.

·       Methodology for Hausman test and Individual Heterogeneity test should be given in Material and Methods section.

·       Keep the short title for Fig 1 and 2.

·       Formatting of the table as per journal guidelines

5. Conclusions: It is noted few policy implications are not from the finding of the manuscript.

Major revision

Author Response

(The authors gave the same response as above.)

Reviewer 3 Report

Dear authors,

thank you very much for an interesting article on farmers decision to participate in a corn to rice project in Northeast China.

Summary 

As mentioned in the title, the authors analysed which factors influence farmers decision to participate in a “corn to rice” project in the Northeast of China. The results are based on 639 answered questionnaires by households in 2019. The results allowed to see, how many households had decided to participate in a corn to rice project and to highlight factors which affected the decision. They used the Probit method to find the important factors affecting to participate in the project. For the households, an improved income is the main driver to switch to rice production, followed by the possibility to have access to family labour. They conclude that farmers will participate in the corn to rice project, as long as they experience a switch from corn to rice production as economic beneficial for them.

The work allows to get an understanding in households decision making in regard to proceed growing corn or to participate in a project to convert from growing corn to grow rice. For the administration, the results allow to target their means when their goal is to increase the area of rice grown.

General concept comments

Somehow, it seems to be obvious to the reader, that the goal of the work is to analyse the factors influencing farmers decision to participate in “corn to rice” projects in the Northeast of China (line 14). This should be stated at the end of the introduction. There, it could also be mentioned if they have any hypotheses, maybe based on the findings from southern regions (line 41).

The authors analyse, which factors are relevant for households to participate in the “corn to rice” project (line 42-44). But for readers outside the Northeast of China, it would be really helpful to get a short description of these projects. Do farmers commit to convert to grow rice when they participate, do they have to convert the entire farm area or only a part? Are they still allowed to grow corn? Do farmers get any payment for participating or higher prices for rice than farmers which do not participate?

The authors use a household survey and present variables. For an agricultural production as corn and rice, I would expect much more variables to be relevant to include to get an understanding of the farm. Important variables should be the area for each farm, share of different crops grown, yields for different crops, amount and cost of inputs as fertilisers, chemicals and fuel, as well as an easy gross margin calculation for corn and rice for the respondents. The authors mention the conversion from corn to rice as a “market economic activity”. In this case, I would expect much more economic data from the farms as you present. It is surprising, that there is no information in the article how much area was converted from corn to rice in the project period, neither as a percentage of the farm or former area with corn. I can only find one indicator, the dummy variable if farmers are participating in the corn to rice project or not. But what does this indicate? Have farmers to convert the entire farm area from growing corn to rice, or just the area for corn? How large is the share of corn and rice in relation to the entire farm area?

Many of the variables used and presented are not sufficient described in regard to what they actually represent, and which units are used. And the variable code is difficult to understand.

Since this is a scientific article and no review article, I would expect that “2. Literature review” should be part of the introduction. After presenting preliminary findings, the goal of the work and possible hypotheses should be presented for the reader.

The last part of the Materials and Methods section includes an interesting discussion. Usually, in this journal, there is an own discussion section. Such a discussion would be beneficial to add to your manuscript. Switching from growing grain to rice can have an important impact on the environment, especially due to the possibility of increased emission of nitrous oxide from farmed area. Publishing in “Sustainability”, this should at least be an important part of the discussion section.

In regard to sustainability, there is just a hint, that rice production with irrigation can give more stable yields. This could increase food security. But the use of irrigation could also give more stable yields for corn, I suppose. Without irrigation, rice production seems to be quite vulnerable. Results on the amount of corn and rice production for farmers participating in the corn to price project are lacking, both in regard of yield on farm and all farmers in the project.

Also, a section “Acknowledgement” with information of different contributions and possible funding should be added.

For the entire text, the English has to be improved.

Specific comments

Below, you find specific comments. To reduce my workload, I don’t mention everything I found. See this feed-back to show how to proceed with the work.

Line 2: “…”Corn to Rice” Farmers…” As I understand, households were asked to participate, regardless they participated to the project or not. You should rephrase the title. Maybe something like “Analysing of Influencing Factors for Farmers in Northeast China to participate in the “Corn to Rice” project”. Maybe better with “Analysing of Influencing Factors for Farmers in Northeast China to convert from Corn to Rice production”. For the readers, I guess, it is of less importance if farmers participate in the project then the decision to produce either corn or rice.

Line 12: You mention a several number of corn to rice projects. Later, you only mention one corn to rice project. Readers outside China probably do not now if these projects differ and need to get some brief information about the main highlights of the project, you refer to.

Line 15-20: You mention “relative benefits” as “improved farmers income” and abundant family labour as important for the decision to convert from corn to rice production. Why do you describe this as “market economic activity”?

Line 21-22: You conclude that “the government should not participate too much”. Isn’t the project initiated by the government? How much are product prices for corn and rice influences by the government?

1. Introduction

Line 26-33: Write if these are data for China or Northwest China. For which period are your findings?

Line 34: You write: “Changes in prices promote the change of corn to rice”. I don’t thin that price changes induce the change, but the price-relation between corn and rice, when prices for rice increase in relation to corn. Or maybe when rice prices may be higher and more stable.

Line 37-38: You write “but also to the change of land quality”. I don’t expect land quality to change (in a short time-period). Pleas check what is mentioned in the article referred to.

Line 41: Are these questions you want to answer?

Line 42-44: Is there a reference for the survey? Are the results published (in English)?

Line 52: You mention here “corn, soybeans and millet”. Thus, it can be expected that farmers grow more than just corn and rice. But other plants are absent in your tables. Why? Can you include more information in the tables?

Line 54-60: This part should be in line with the final structure of this manuscript.

Line 60: If you write “basic conclusions and policy suggestions”, the reader should get more information about the corn to rice project to understand what the project is about and why you give your conclusions.

2. Literature review

Merge this part with the introduction.

Line 66: change “country” to “China”.

Line 74-75: If rice contributed by 21% of the overall increase in grain-production, it can be expected that other plants contributed more to this increase. This opens the question why it is a goal to increase the growing of rice?

Line 81: Can not understand the part: “economic benefits are the core demands”.

Line 88: You write “agricultural inputs have decreased”. What type of inputs, fertilisers, chemical, fuel, machinery, handwork? Mentioning inputs here, it is important to relate income from sold products to cost from inputs to see the resulting gross margin. This would be really important for the reader to understand farmers decision but is absent in the manuscript. Please add this information.

Line 92-93: Writing “room for improvement in the utilization rate of agricultural machinery”, it is surprising, that there are no detailed questions about machinery in the survey, especially knowing that irrigation can be crucial for rice.

Line 101: Writing “operating scale of each household is less than 3.33 hectares”: can you add the size of the farms in the questionnaire? The size may impact farmers decision. In addition, it would be important to know how much of the farm area is used for corn, how much for rice and how much for other plants. What about animal husbandry?

Line 106: Writing “gradually concentrate land on farming experts”, it would be beneficial to know about agricultural education among the respondents. This can have an impact on the decision to grow rice.

Line 114-115: “Based on existing research, this paper will analyze the factors that influence farmers to switch from corn to rice.” This could be the goal of this paper, but then it would be more common to base your analyses on your results and the survey instead of “existing research”.

3.Materials and Methods

Line 125 and 127: Be clear when you use “benefit” and when “net income”. Looking on information in the tables, it seems, that you do not have information about “net income”, only “income level position in village”, planting income and subsidies.

Line 136-137: You focus on “the panel binary selection model that affects farmers' participation in the corn-to-rice project”. But still, the reader does not know enough about this project and the impact on farming. How much area do farmers have to convert to growing rice? All variables used have to be clearly described.

Line 144: “planting income of the land respectively”: When you want to do an economic evaluation, you do not need only the income, you also need the variable costs for inputs used. Especially, when growing corn and rice can have different use of inputs.

Line 145: You mention “proportion of household agricultural income”. Probably, this refers to “income from land cultivation”. But is this correct? The first one could also include income from animal husbandry.

Line 147: “whether they have part-time work experience” – in Table 1, I only find “Do you have any working ex-perience”. Shal this be the same? For me, it is different.

Line 147: “whether the local government provides relevant training” – what is “relevant”? How to grow rice?

Line 149: “whether to join cooperatives” – what type of cooperatives? Agricultural?

Line 163: Change from “very demanding on the respondents” to “very demanding for the respondents”

Line 163: What do you mean by “Therefore”? Because of missing or inconsistent data?

Line 163: What are “sample estimates”?

Table 1:

Try to find “variable codes” which are quite close to the “variable name”. E.g. “education” for “education level” is easy to grasp. “drywat” for “Change from corn to rice” is challenging for the reader.

For some of the variables you must list the unit. E.g. “Amount of subsidy for rice cultivation per mu”: are the subsidies in yuan? What unit is mu, an area? Can you give a footnote for number of square meters?

Insert area of the farm.

Insert data for animal husbandry. This can highly affect farm income and have an impact on the workload.

Information on other plant products, especially vegetables, demanding mor work and possibly contribute to a higher income.

“working experience” – what type of working experience? In agriculture, in growing rice…?

“Income level position in the village” – is this income from agriculture or all income? Should they rate their income level or do they know the income of others?

“household consumption expenditure” – given in yuan?

Is there any information about self-sufficiency from farming? Can this be related to “household consumption expenditure”?

“Does the local government provide relevant training?” – what type of training? Agricultural, growing rice any other?

Are the subsidies for rice and corn differing on differing work to do, using the different wordings “rice cultivation” and “corn planting”? Or are the different subsidies paid in regard of the area with the different crops?

“Whether to join a cooperative” – do they ask for the possibility or if they wish to join?

“Are there irrigation facilities near the cornfield?” – The answer is of little help as long as there is no information if they can get access to the water.

Line 168: You start here to present findings from Table 2. This is fine. In Table 2, you use the share, 0.54 of the respondents “carried out corn-to-rice projects”. But in the text, you change to percent, mentioning “54.46% of farmers in Northeast China”. It would be better for the reader to use the same unit, either share og percent and in addition the same values, either 54% or 54.46%. In addition, the result refers to the respondents. Not “54.46% of farmers in Northeast China” but “54.46% of the respondents have carried out corn-to-rice projects”. Later, you can discuss how representative your findings are.

Line 169: Why do you mention, that “the proportion of corn-to-rice is relatively high”? Is this compared with other findings?

Line 170: You write “The average income per mu of farming is 1,038.57 yuan, but there are certain fluctuations ranging from 225 yuan to 2,240 yuan.” I can not find this information in the table. I would expect this as e result from “lnbenefit”, divided by the area of the farm in mu. Is this variation between the participants? Or do you relate for the average of farms for the three years?

Line 171-172: You write “It is this fluctuation that causes farmers to be willing to change corn into rice.” Yes, there is a high variation in income per mu of farming. But it should be crucial to know the reason for these differences, in can be the size of the farm, it can be a different share of different crops, with vegetables giving higher income, it may be due to animal husbandry. In addition, the farm income is giving little information without the costs for inputs. At least, there should be a simple calculation of cross margin.

Line 172-173: You mention “rice planting process relies more on agricultural machinery irrigation”. This underlines the importance to include some simple calculations of cross margin. When growing rice depends on use of additional machinery, this is a type of investment. Maybe many farmers have not the ability to pay for irrigation, despite this would allow to get more stable yields.

Line 175-176: Yes, yields depend on weather conditions. Would it be possible to get more stable yields, when irrigation would be used for corn? Then, also the yield from corn would be more predictable.

Line 177-180: Adding the mentioned percentage of education, sums up to 114 % with education. This must be corrected.

Line 179-180: Correct the sentence: “accounting for 27%. ratio of 4%.”

Line 180-181: What do you want to describe with the “overall proportion” in the sentence: “Farmers with part-time work experience accounted for 40.53%, and the overall proportion is not high.”

Line 182-183: For me it is not possible to understand the sentence: “With the increase in the income level of migrant workers, farmers in the Northeast region rely more on land for income growth, and their income is relatively weak.” How is the income of migrant workers and farmers related? Have these migrant workers in earlier times been working for less money on farms? Would it ne an alternative for farmers to increase income by producing vegetables or increasing animal husbandry?

Line 184: Here, it is the first time, that you specify, that this training is for farmers. Why not mentioning this before, using the unspecified term “relevant training” without mentioning that it is relevant for farming. But you ask if they get training. This implies now, not earlier. So, many can have had agricultural training before, but you do not get an answer on it by the survey.

Line 186: What are the “three rural issues”? This maybe not clear for readers outside China.

Table 2 “Basic information of the main variables of the model” – reconsider the title. Would it be better to mention the table as “results from the survey”?

Improve the variable names as mentioned before and add a column with the unit for each line. Write dummy, where they are used. You can skip the number of observations, because the number is identical and write this number only once as additional information below the table. The numbers used should be identical with the once used in the text.

Table 3

Here you use the word maize, not corn. Do you want to highlight something by this?

As underlined before, it is still not clear when you group farmers to belong to the group “did not change corn to rice” and “Change corn to rice”. Is the criteria solely to participate in the project? Do farmers have monoculture with corn and do they converted the entire area to rice? This is important to understand the decision-making of the farmers.

Why doe you write “not a village cadre” and in comparison, “is the village cadre”? Why not “is a village cadre”?

Why is there only one line for irrigation “gender; cornirr”? Is there really no women using irrigation? Maybe good to add as gender “male” for the line.

Line 193-227: Here you present an interesting discussion. But you are in the section for material and methods. Move the discussion to a discussion section.

Line 202-204: You write “the proportion of village cadres who carry out corn-to-rice projects is about 16%, and the proportion of farmers who are not village cadres is as high as 47%.” Based on the number from Table 3, I cannot understand, how get the numbers 16% and 47%.

Line 217: You write “respondents who have no working experience tend to make a living by farming”. Do you evaluate farming as “no working experience”? I disagree!

Line 219: You write “while the income from rice tends to be higher”. Again, a simple gross margin calculation would be important to underline your statement. A higher income is not worth much, when producing costs increase.

Line 225-226: You write “that the government's training will only help farmers better cultivate good land”. Do you want to express that training only helps on good land or to cultivate land in a good way?

4. Research Results

Line 229-238: I would prefer that you start presenting the results for each model one by one before you argue to select the best one.

Line 250-251: Rephrase “that the variables used in this paper do not have serious collinearity problems”. The variables have no problem with collinearity! But if there is no collinearity, we skip to handle this.

Line 256: “The benefit after changing corn to rice has a significant positive impact on farmers' decision to change corn to rice” If you ask farmers about a decision, they expect a benefit, it is not given, that they will reach this benefit. And again, be clear how you use benefit. This should be more than just economic outcome.

Line 260-262: “The greater the benefit of the corn-to-rice project, the more motivated they will be to do the corn-to-rice program, and the more likely they will be to do the corn-to-rice program.” This is new for the reader outside China. What is the difference between the corn-to-rice project and corn-to-rice program? It seems to be important to differentiate. And again, what do you include in the meaning of “benefit”?

Table 4: As mentioned before: improve the variable names.

You write about model 2, 3, and 4. In the table, you use OLS, logit, rlogit, and progit. Do it easier for the reader.

Line 267-286: Interesting discussions. But it should be clear, where you present and where you discuss.

Line 284: “…only labor can be hired to complete the work, which will obviously greatly reduce the profit level of agricultural cultivation” – here you go in the direction of gross margin. And it is difficult that you assume, that household members work for free. This will depend on the fact if they have other paid work or no other work.

Line 289: Write the formula in a separate line.

Line 322: Change “can effectively analyze the survey data” to describe”.

4.3.4. The influence of individual characteristics on the behaviour of changing maize to rice

The number of family members is not an individual characteristic in my eyes.

Figure 1 and 2. For serv = 1 and workout = 1, you should not use a circle. Use for example a triangle.

After line 338: I would suggest to gather all discussion in a separate chapter.

Line 341: You write “it is found that whether farmers participate in the corn-to-rice project is mainly affected by the benefits”. And in the next sentence, the only benefit mentioned is an increase of agricultural income. Why not write that this seems to be the main driver?

Line 357: Fine, that you mention here the profitability! But where is more information than just focus on income without information about the costs?

Line 359-360: You write “the project of changing corn to rice requires a higher number of farmers' families”. I guess, you think of “higher number of farm-family members”. But his assumes that family-members are cheaper than other workers.

Line 361-362: You write “the government should recognize that the corn-to-rice project is a market economy activity”. As before, give the reader a short introduction about the corn-to-rice project and the corn-to-rice program.

Line 365: Improve the sentence: “As a functional department, the government should…”. The government is not a functional department.

Author Response

(The authors gave the same response as above.)

Round 2

Reviewer 2 Report

The authors incorporate all the suggestions and this manuscript is recommended for the publication

Author Response

Thank you so much for the help. We have made some modifications based on the editor's suggestion. 

Reviewer 3 Report

Dear authors,

Thank you very much for the detailed answering of my comments and suggestions.

You have been working hard to improve the manuscript and it was a pleasure to read the result.

Please find detailed comments and suggestions attached as a PDF-file to improve your manuscript further.

With the goal to publish in “Sustainability”, I still suffer a comment regarding growing rice instead of corn. You mention that there is a goal to produce more rice in China. But are there publications on the environmental effects, you can mention and discuss, as for example about emission of climate gases and nutrient runoff?

Thank you for the detailed reply on my questions. Often, I asked them because I expect that many readers will have the same question. Thus, it is good if you consider including important parts of your answers into the text.

Reconsider specially to include some parts of there replies in the manuscript:

Reply, 38, 39, 48 for “migrant workers”, 66: formula still within the text and difficult to read, and 69 colour changed, but still both as circle.

Kind regards

Author Response

Dear Editor :

Every comment you give me is very detailed, and I really, really appreciate it. These suggestions are some of the best I've seen. These suggestions can effectively help me to modify the paper, which is of great help to me. We have made the following modifications according to your suggestion. 

Thank you again for your help.

Best regards,

Luan Wang
